# Retired Athletes and the Intersection of Food and Body: A Systematic Literature Review Exploring Compensatory Behaviours and Body Change

**DOI:** 10.3390/nu11061395

**Published:** 2019-06-21

**Authors:** Georgina L. Buckley, Linden E. Hall, Annie-Claude M. Lassemillante, Kathryn E. Ackerman, Regina Belski

**Affiliations:** 1Department of Health Professions, Swinburne University of Technology, Hawthorn, VIC 3122, Australia; alassemillante@swin.edu.au (A.-C.M.L.); rbelski@swin.edu.au (R.B.); 2Consumer Representative, Victorian Institute of Sport, Albert Park, VIC 3206, Australia; linden.run@gmail.com; 3Division of Sports Medicine, Boston Children’s Hospital, Boston, MA 02115, USA; kathryn.ackerman@childrens.harvard.edu.au; 4Neuroendocrine Unit, Massachusetts General Hospital, Boston, MA 02114, USA; 5Harvard Medical School, Boston, MA 02115, USA

**Keywords:** disordered eating, body image, binge eating, sport, retirement

## Abstract

Background: Retirement from elite sport is a unique transition that influences significant identity, body, and lifestyle changes. This mixed-studies systematic literature review reports on athletic retirement, maladaptive eating behaviours, and body dissatisfaction. Methods: The preferred reporting items for systematic reviews and meta-analyses (PRISMA) guidelines were followed to search the following databases: Web of Science, Scopus, PubMed, EBSCO Host, Sport Discus, and CINAHL. Sixteen studies were synthesised and contrasted through thematic analysis to develop three overarching themes. Results: The three themes that arose include body dissatisfaction and grief, disordered eating and compensation, and long term influence of sporting culture. Maladaptive and compensatory behaviours can arise from sustained athletic identity, body grief, lack of education, and contradictory body ideals. Conclusion: The concept Athletic Body Transition is defined as exploring how a lack of body acceptance may lead to maladaptive behaviours related to food, exercise, and body arising in this transitory period. This review identifies the need for sporting organisations and health professionals to acknowledge this significant transition in regards to athletes’ relationship with food and body subsequent to a sporting career.

## 1. Introduction

Athletic retirement is a unique occupational transition associated with a plethora of changing behaviours related to nutrition, body, and lifestyle [1]. It is experienced only by those who engage in sport at an elite and, occasionally, sub-elite level. An athletic occupation encompasses more than just an individual’s time, extending to the broader lifestyle and identity of an individual beyond a traditional career [1]. Dependent on the sport, athletic retirement is often characterised by a reduction in physical training causing loss of lean muscle mass and fitness, altered body composition, and detraining or changes in nutritional practices as a function of occupation [2]. ‘Flourishing’ or successful transition to athletic retirement requires consideration of many factors [3]. The reason or cause for retirement and the ability to emotionally adapt have been conceptualised as factors relating to successful athletic retirement [4]. Reason for retirement, personal development, career achievement, social support, and education have a significant effect on the early stages of transition [5,6]. Most notably, continued athletic identity has a significant negative association with retirement outcomes [7,8]. Continued athletic identity occurs when individuals continue to perceive themselves as ‘athletes’ after retirement, which affects self-perception, beliefs, qualities, and their worldviews [9]. The current system of athletic support largely neglects what happens to athletes after they cease to compete, often ignoring the prolonged influence of sport on their lives [4]. 

Nutrition and body composition are functions of an athlete’s occupation, sometimes creating a complex relationship with food and body for the individual. Athletic retirement is a disruption to this already complex relationship that provides an environment to precipitate compensatory behaviours [10]. The landscape of disordered eating and eating disorders has significantly changed with the introduction of the Diagnostic and Statistical Manual 5 (DSM-5) in 2013 [11]. The diagnostic definition of eating disorders was broadened with the introduction of binge eating disorder and the removal of amenorrhea from the anorexia nervosa criteria, resulting in reduced gender-based stigma and treatment barriers [12]. Further research of muscle dysmorphic disorder, compulsive exercise, and orthorexia nervosa are expanding the perception of what is constituted as maladaptive in relation to food and body [13,14]. Compulsive exercise, also known as exercise addiction, is defined as a compulsive obsession to exercise, often pushing through injury and illness [15]. The introduction of these terms reinforces the intersection of the relationship between food and body and the complex behaviours that ensue. We have yet to fully understand how these constructs relate to the athlete population and the further, vulnerable, at risk sub-group of retired athletes. 

The primary objective of this systematic literature review was to compile and appraise the evidence of qualitative, quantitative, and mixed-methodology studies pertaining to the relationship individuals have with food and their bodies after retiring from elite and sub-elite competitive sport. This includes studies exploring the influence of athletic identity, body monitoring, sporting subculture, expectations, athletic body idealisation, body composition, and relationship with food on an individual’s life as a result of their participation in sport. Exploring the psychopathology and clinical diagnostics of retired athletes extends beyond the scope of this paper. The preferred reporting items for systematic reviews and meta-analyses (PRISMA) [16] were used to answer the following research question: how does athletic retirement affect an individual’s relationship with food and their body? This review also aimed to summarise the current scholarly limitations and make recommendations for future research developments and sporting sector improvement. 

## 2. Methods

### 2.1. Search Strategy 

A systematic mixed-methods literature review was conducted using the preferred reporting items for systematic reviews and meta-analyses (PRISMA) guidelines [16]. The review was registered with PROSPERO (International Prospective Register of Systematic Reviews) and, as such, the protocol can be found via the following registration number: CRD42018106470. The PICOS (Population, intervention, comparators, outcome & study design) criteria are addressed in Table 1 to define the research question and synthesis. The following search terms were used: retir*, former and athlete*, sport and eat*, or feed*, food or disorder*, anorexi*, bulimi*, weight, ‘body image’, or ‘body dissatisfaction’. The review aims to primarily describe dietary behaviours and outcomes and, as such, studies referring exclusively to body were excluded. All eligible peer-reviewed journals published before August 2018 in the following databases were included: Web of Science, Scopus, PubMed, EBSCO Host, Sport Discus, and CINAHL. To minimise bias, two independent reviewers (GB and LH) screened the titles and abstracts and conducted full text analyses. Any discrepancies between reviewers were discussed with a third independent reviewer. Inclusion criteria were: (1) English language peer-reviewed journals; (2) studies with participants including male or female retired athletes; (3) studies referring to either (a) eating behaviours with or without (b) body image/body dissatisfaction. The outcomes assessed indicated: disordered eating, eating disorders, maladaptive food behaviours, restriction, rigid food control, body image issues, weight control, body control, compensatory exercise, negative self-image, and dysmorphia. Studies were excluded if: (1) the full text was not available, (2) the only reference made to food or body occurred from the lens of retired athletes using retrospection of their athletic career, or (3) the aforementioned outcomes were not included in the study results. Due to a lack of quantitative study homogeneity, a meta-analysis could not be conducted. Therefore, all eligible qualitative, quantitative, and mixed-method studies were synthesised. The bibliographies of the selected studies from database searches were mined to identify additional studies that met the inclusion criteria.

### 2.2. Data Synthesis and Analysis

Eligible texts were synthesised independently by two reviewers (GB and LH) with a third and fourth reviewer (AL and RB) available to negotiate discrepancies. The final list of included studies was categorised based on research methodology (qualitative, mixed-methods, or quantitative). Quantitative studies were summarised, contrasted, and compared while qualitative studies were thematically analysed and synthesised. The mixed-methods results were both thematically analysed and quantitatively compared and contrasted (see Table 2, Table 3 and Table 4 for a summary of eligible studies). The evidence of all studies was grouped according to common themes. To minimise bias, piloted forms and coding strategies were implemented by GB and LH. Thematic analysis was discussed and agreed upon by all four independent reviewers. Study quality was appraised by GB and LH using a scoring system for mixed-method research and mixed-studies review [17], with 100% being the highest score. Summarised percentages can be found in Table 2, Table 3 and Table 4. 

## 3. Results

### 3.1. Eligible Studies and Thematic Development

Three hundred and thirty eight studies were identified from select electronic databases. Sixteen of these studies met the inclusion criteria and are included in this systematic review. Reasons for exclusion are outlined in the PRISMA flowchart (Figure 1) [16]. These studies were published between 1996 and 2018, and included eight cross-sectional or cohort studies, four mixed-methodology studies, and four qualitative studies. The studies are presented under the three key themes identified in the review as follows: ‘body dissatisfaction and grief’, ‘disordered eating and compensation’, and ‘long term influence of sporting culture’. 

### 3.2. Body Dissatisfaction and Grief

Nine of the 16 studies presented findings that fit the theme ‘body dissatisfaction and grief’ [23,24,25,26,27,29,30,31,33]. Of the nine studies, three were quantitative [23,24,25], three were mixed-methods [26,27,29], and three were qualitative studies [30,31,33]. This theme encompassed studies that described how body dissatisfaction was related to the construct of body idealisation and the changing body in athletic retirement. This theme was most evident in studies exploring the first few months to years of retirement, especially when significant body composition changes were identified by the individuals [29,31]. 

Studies fitting the theme ‘body dissatisfaction and grief’ identified how, in Western society, the changing body can be a risk factor for maladaptive eating and exercise behaviours [27,33], and explored how athletic identity influences perception and acceptance of change [26,29,33].

The studies highlighted that body dissatisfaction can arise from the change in composition independent of an increase in weight or body mass index (BMI) [23,27]. The study by Marquet et al. [23] demonstrated that retired weight-class athletes gained less weight as they aged compared to age matched controls (BMI: 3.2 kg/m^2^ vs. 4.2 kg/m^2^) up to 50 years after retirement, despite retired athletes describing heightened levels of body dissatisfaction. Furthermore Papathomas et al. [27] demonstrated that even though 75% (*n* = 162) of retired gymnasts and swimmers were classified in the ‘healthy weight range’ (BMI: 18–25 kg/m^2^), 55% were still dissatisfied with their current body and a further 60% were engaging in weight loss practices.

Another five studies identified the importance of the passage of time after retirement [23,28,29,31,33]. Overall, the studies suggested that time after retirement appeared to alleviate body grief and influence the movement towards body acceptance. However, the first few months following retirement was established to be challenging. For example, Stephan and Bilard [29] showed that after 4.5 months of retirement, body image outcomes were worse than after 1.5 months. As time from retirement increased, the literature demonstrated changes in the reverse direction, with athletes who had been retired for more than five years appearing to have better relationships with their changing body versus athletes retired for less than five years [29,31,32,33]. This appeared to improve even further with more time out of the sport. O’Connor et al. [24] showed that 15 years after retirement former athletes had lower scores of body dissatisfaction than age-matched controls. Papathomas et al. [27] proposed that body dissatisfaction was independent of time and more closely related to the influence of sporting subculture and body monitoring. Overall this suggests that whilst body acceptance is more likely with time, other confounding factors continue to influence an individual’s relationship with their body.

The retired athletes who noted increased adiposity and noticeable body composition change were those who experienced higher levels of body dissatisfaction [26,27,29]. This body dissatisfaction was described in the qualitative papers as body grief or reported as loss of a former self [26,27,29,33]. A swimmer, in the study by Greenleaf [26] recalled, “I haven’t fully accepted it for what it is” when referring to her body composition changes over a year after retirement. Athletes who held onto a strong sense of athletic identity were more likely to commit to a former self and disconnect from their organic body changes [26,27,30,31,33]. This body dissatisfaction, body grief, and negative affect were described to have broader effects on self-esteem, self-worth, physical condition, competence, and attractiveness [25]. 

### 3.3. Disordered Eating and Compensation

Eleven out of 16 studies presented findings that fit the theme ‘disordered eating and compensation’ [18,19,20,21,22,27,29,30,31,32,33]. This theme encompassed how unique stressors in the transition from the elite sporting environment to athletic retirement triggered maladaptive behaviours such as disordered eating or compensatory exercise. This theme was most evident in studies of retired athletes who formerly participated in sports with high energy consumption, i.e., endurance sports [30,33]. 

Unrealistic goals of body composition maintenance were associated with established restrictive behaviours [27,29,30,33]. “Unwanted body changes” in participants were reported as triggers for compensatory exercise and meal skipping with subsequent binge eating [30]. Compulsive exercise, also known as exercise addiction, was used as an additional compensatory behaviour developed in retirement as a means to control body dissatisfaction and energy balance [33]. Retired athletes who had competed in high energy consuming sports were often the ones to engage in compulsive exercise to match the energy output of their former exercise levels [33].

Disordered eating behaviours, such as binge eating and dietary restriction, were frequently referred to throughout the qualitative literature [27,28,30,32,33]. Retired swimmers reported that when they skipped workouts they felt the desire to skip meals to compensate [30]. This dietary restriction led to further restriction, compensatory exercise, or binge eating behaviours. Often participants described eating ‘forbidden foods’ and ‘eating everything’ upon retirement [30,33]. Adverse nutrition behaviours, defined as ‘irregular meal patterns and infrequent nutritious foods’, were explored in four quantitative studies [18,19,20,21,23,24,25,26,27,28,29,31,32,33]. These studies reported that 42%–65% of retired athletes engaged in such adverse nutrition behaviours, higher than the reported 26% of currently competing athletes. Disordered eating was measured by Marquet et al. [23] and O’Connor et al. [24], who found insignificant differences in disordered eating between retired athletes compared to age-matched controls. Kerr et al. [22] reported uncontrolled eating in 5.8% of the retired athlete population, which is higher than in the general population. In a qualitative study, Kerr and Dacyshyn [31] described participants with eating disorders persisting through their athletic career to retirement. However, for some retired athletes, disordered eating behaviours ceased, as suggested by O’Connor et al. [24], who compared retired athletes to the general population and saw lower subscale scores in ‘drive for thinness’ (athlete: 3.3 ± 5.3, control: 4.0 ± 5.3) and bulimic symptomatology (athlete: 0.6 ± 1.1, control: 1.0 ± 3.0). 

### 3.4. Long Term Influence of Sporting Culture

Eight out of the 16 studies presented findings that fit the theme ‘long-term influence of sporting culture’ [18,19,21,26,27,29,31,33]. Of these, three were quantitative studies [18,19,21], three were mixed–methods studies [26,27,29], and two were qualitative studies [31,33]. This theme encompassed studies that described how participation in the sporting environment affected the individual after retirement. The long-term and problematic effects reported in these studies were related to continuing athletic identity and past difficult experiences during the athletic career.

Gouttebarge et al. [19] reported that athletes who experienced career dissatisfaction, career injuries or surgery, and inadequate social support were 2.4 times more likely to develop symptoms of eating disorders in athletic retirement. In retired elite rugby players and footballers, stressful life events and career dissatisfaction increased the risk of adverse nutrition behaviours [18,21]. 

Individuals who continued to identify as athletes had a poorer relationship with food and body [7,26,27,29,31,34]. Athletes who continued to engage in their athletic environments experienced heightened levels of objectification and were held back in their ability to re-identify [33]. Continued involvement in the sport meant that those around them still perceived them as an athlete. “I really enjoyed coaching, but they [the other coaches] were still really negative about my weight and shape” [33].

Kerr and Dacyshyn [31] proposed that those who struggled with food and body issues in athletic careers had even more difficulty transitioning to athletic retirement. Retired athletes often described heightened levels of objectification from increased body monitoring and awareness, a trait developed in their sports [26,29,33]. Runners in the study by Greenleaf [26] noted how their running shirts (crop tops) heightened objectification of their bodies and still influenced their body image one to five years after retirement. Another runner noted how difficult she found it to stop comparing her body to other people, a skill learned in the sporting culture of competition, perfectionism, and comparison [26]. Continued worshipping of athletic body ideals was a further setback for body acceptance in retired athletes, where, for example, former gymnasts continued to perceive “thinness is a measure of success” [33]. 

## 4. Discussion

The key drivers of the poor relationship with food and body in retired athletes synthesised throughout this review included (1) continuing athletic identity, (2) establishing unrealistic expectations related to nutritional and body composition changes, (3) athletes with significant energy balance changes or significant body composition shifts away from societal body ideals, and 4) the reason for and length of time since retirement. These key drivers were highlighted as the most common and pertinent factors across all appraised studies. As such, the body composition changes that occur in athletic retirement can be likened to other significant times of body change, such as puberty, pregnancy, and menopause. It is often in these times of body change that people are at heightened risk for disordered eating [35].

Many athletes are known to under-fuel, causing significant physiological and performance detriments as described by relative energy deficiency in sport (RED-S) [36,37]. There is strong evidence that athletes have higher eating disorder rates than the general population [38]; however, there is less evidence on how we can support these athletes and the sporting cultures to which they are exposed. We also know little about how these athletes assimilate back into regular exercise and eating patterns beyond their elite careers and into athletic retirement. 

The landscape of eating disorders and body image has changed with the redefinition of DSM-5 eating disorder features and the rise of muscle dysmorphia, compulsive exercise, and orthorexia nervosa as constructs [11,13,15]. Most articles synthesised in this review were written prior to this change and often described problematic behaviours aligned with these terms but failed to label them. Binge eating disorder was introduced in the DSM-5, but has not been widely considered in the literature, especially in athletes, possibly due to a lack of accurate and validated assessment tools or prevalence exploration. It is characterised by regular uncontrolled binge episodes with subsequent guilt, shame, and further compensatory restriction [11]. Orthorexia nervosa is a term characterised by a preoccupation and obsession with ‘health’-related foods or behaviours [13]. These terms best describe many of the qualitative retirement experiences, but have not been fully characterised or quantitatively assessed due to the recent emergence of the terminology. 

Athletic ‘body ideals’ are body standards governed by gender and sporting subculture. They can align or contradict broader societal idealisation of thinness for women and muscularity for men [39]. These social or athletic ideals cause significant weight or shape concerns in individuals and have been extensively linked to eating disorder development [35]. The athletic ideals of individuals who identify as athletes are more influential than their social body ideals in shaping their relationship with the body. For example, males who identify as marathon runners are known to emaciate themselves to meet athletic thin ideals, despite the popularised male muscular ideal [40]. Upon retirement, a movement away from athletic identity ensues with an increased influence of social body standards, however not without a paradoxical transition. 

To conceptualise this complexity of body acceptance and influencing factors identified in this review, we have proposed the concept ‘athletic body transition’ (Figure 2). The term ‘athletic body transition’ is best described as the body image or body dissatisfaction a retired athlete experiences based on a number of active and passive factors. The factors most likely to influence this include: sporting culture, former athletic body ideal pressure, body composition genetics, retirement body composition changes, and level of continued athletic identity. This broadens the former definition by Papathomas et al. [27] of the ‘retired female athlete paradox’ to include the influence of athletic and gender-based societal body idealisation, identity, body grief, and body composition changes. For example, female swimmers have an ideal muscular somatotype in their sport and yet are exposed to the societal thin ideal. In retirement, when muscle mass is lost, body acceptance can be gained as a movement towards female ideals occur. Alternatively, female distance runners, whose sporting ideal transposes that of the social ‘thin ideal’, may have decreased body acceptance upon retirement due to the movement away from body sporting and social ideals with lost lean muscle mass and increased fat mass. This concept is different to body dysmorphic disorder as it is a unique transition known to athletes having acute body composition changes and, in some instances, changing between body ideals. This process is often transient, largely subclinical and is not defined by an individual seeing their body differently, rather it has to do with the acceptance of body changes.

Continued athletic identity with body composition changes may further reduce the chance of body acceptance. Identity is made complex in athletic retirement by athletes objectively continuing to see themselves as athletes, in some cases, long after they have ceased competitive sport [10]. For instance, if a former athlete looks in the mirror and continues to see an elite athlete, there is more likely to be dissatisfaction with the natural body composition changes, reducing the chance of body acceptance. Moving forward is increasingly difficult for former athletes in the media and the scrutinising public eye. Much information in newspapers, television, and social media presents retired athletes’ bodies unfavourably, as if they are expected to have the physique they had as full time athletes [26,41]. For many athletes, the social support and occupational skills they have are related to their sport, adding an increased layer of complexity in retirement outcomes. Developing social networks and skills during athletic careers is essential for improving identity transitions. Continued athletic identity is a significant barrier to positive food and body outcomes as described in the model of ‘athletic body transition’ (Figure 2).

From the qualitative and mixed-methods studies it was conveyed that body acceptance was made increasingly difficult for athletes who experienced grief or loss of their athletic body as a result of inadequate or unrealistic redefining of identity or body change expectations [26,27,33]. Decreased body acceptance from the change associated with athletic retirement was proposed to increase negative affect and provide a risk factor for maladaptive behaviours to occur, such as controlled eating or exercise motivated by aesthetics. Knowledge, acknowledgement, and education about the factors that influence body image are required to assist the positive transition of retired athletes, and to establish realistic expectations. Compensatory exercise and restrictive eating have been conceptualised to demonstrate how athletic retirement, body composition changes, and body acceptance can influence maladaptive behaviours (Figure 3). Figure 3 has been adapted from Fairburn, Cooper, and Shafran’s [42] conceptualisation of cyclic binge eating disorder behaviours. 

Health professionals, such as dietitians, psychologists, and exercise physiologists, are integral to providing a sense of guidance and framework in the months immediately following retirement. This ensures a level of nutritional, body image, and exercise counselling to help the athlete navigate the often significant change in energy availability. This change in energy availability causes compensation through exercise and/or food or body composition changes. Intuitive eating is an evidence-based strategy for listening and responding to hunger cues and has been described in retired athletes as a means for positive nutritional transition [32]. Intuitive eating acknowledges ever-changing hunger and fullness cues and gives a framework for athletes to regain a sense of control and active skill development. Follow-up after this acute period is essential to further transition athletes to an achievable long-term nutrition and movement pattern. 

Health professionals who engage with retired athletes should have an awareness of the types of compensatory behaviours to which the individuals are prone. Having an understanding of binge eating disorder, compulsive exercise, and orthorexia nervosa is imperative to the identification and early prevention or treatment of this group. The most prominent behaviours are described in Figure 3 and include: compulsive exercise and/or restrictive eating driving binge eating behaviours. This review demonstrates the need for ensuring relevant health care professionals are adequately prepared to assist retiring athletes to prevent or manage these behaviours long term. 

This review highlights the need for further exploration and understanding of eating disorders, body dissatisfaction, compensatory behaviours, and long-term energy availability outcomes in retired athletes. The past studies identified in this review lacked both longitudinal and validated assessment of disordered eating, body image, and body dissatisfaction, demonstrating the need for further research. A limitation of the previous literature was that gender was not often differentiated in outcome measures. Future studies need to be able to differentiate between gender as compensatory behaviours have been indicated to vary widely. As such, we invite the exploration of the hypothesis that retired athletes are at a heightened risk of developing maladaptive behaviours related to food and body through this transitory period. Furthermore, to capture the quantitative data of eating disorders in retired athletes, we need to continue to evolve the construct of maladaptive eating behaviours in athletes.

There are limited reliable and validated screening tools in athletes measuring disordered eating or body dissatisfaction. The evidence, to date, has prioritised anorexia nervosa in clinical female cohorts and the majority of available assessment tools are orientated to this population. Discussions in the literature of muscle dysmorphia, orthorexia nervosa, and ‘drive for leanness’ expanded our understanding of problematic or maladaptive food and exercise behaviours. However, the clinical application of these constructs has yet to be well-established, especially in the higher risk athlete or retired athlete populations.

Athletic retirement is a complex life transition, as it involves the intersection of food and body significantly more than traditional retirement. We concluded that the scholarly understanding of the intersection of food and body needs to be expanded further, especially in athletic and retired athletic cohorts. It is proposed that athletes are potentially at heightened risk of developing disordered, maladaptive, or compensatory behaviours related to changes in this transitory period of their life. This review highlights the need for increased longitudinal and validated data in the retired athlete population. Furthermore, there is an increased need for acknowledgement and support from sporting organisations, health professionals, and society, to embrace change and provide an environment to improve a retired athlete’s relationship with food and body.

## Figures and Tables

**Figure 1 nutrients-11-01395-f001:**
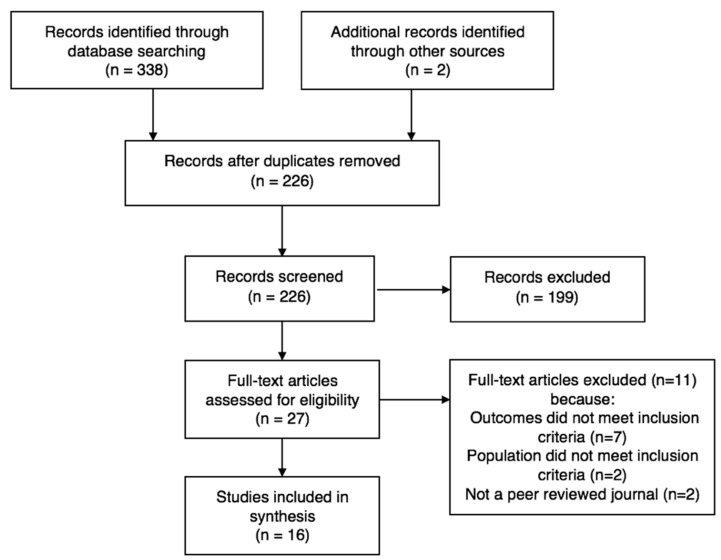
PRISMA flowchart.

**Figure 2 nutrients-11-01395-f002:**
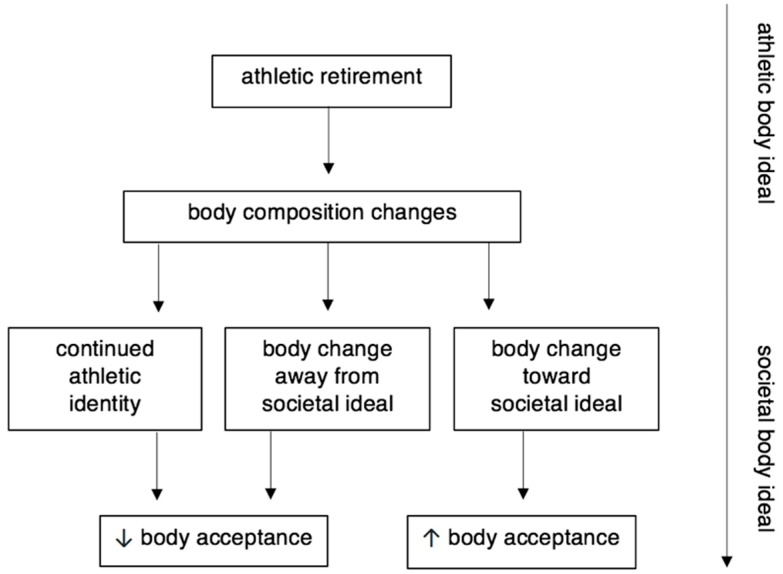
Athletic body transition—a diagram summarising how athletic retirement and changing body idealisation narratives can influence body acceptance outcomes (↓ = decrease, ↑ = increase).

**Figure 3 nutrients-11-01395-f003:**
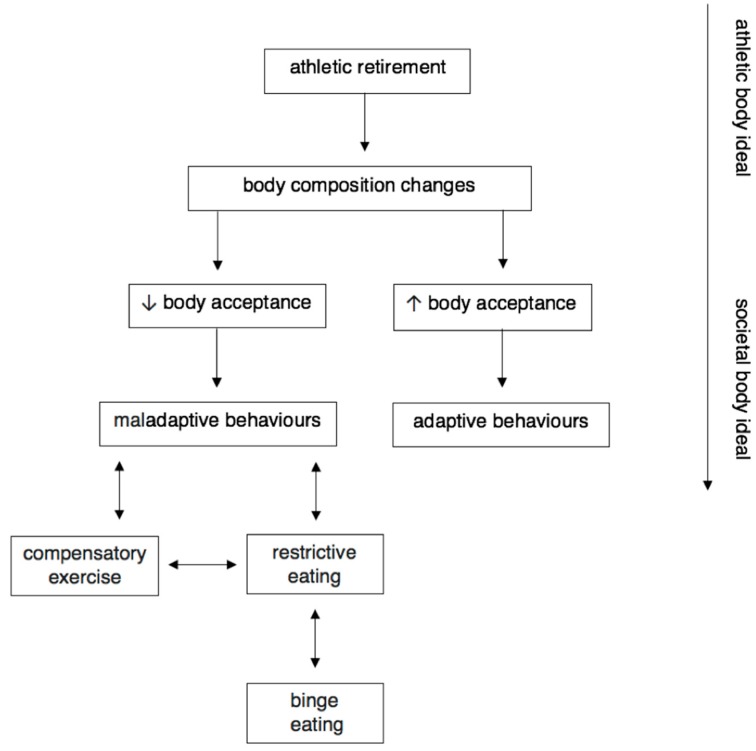
Compensatory eating and exercise in athletic retirement—a diagram summarising how athletic retirement can influence body acceptance and maladaptive exercise or eating behaviours (↓ = decrease, ↑ = increase).

**Table 1 nutrients-11-01395-t001:** PICOS criteria.

Population	Retired Athletes
Intervention/exposure of interest	Relationship with food and/or body
Comparators	NA
Outcome	Primary outcome: disordered eating, eating disorders, maladaptive food behaviours, restriction, rigid food control Secondary outcome: body image issues, weight control, body control, compensatory exercise, negative self-image, and dysmorphia
Study design	All qualitative, quantitative and mixed-methodology studies

**Table 2 nutrients-11-01395-t002:** Eligible quantitative studies.

Author, Year	Country	Participant Characteristics, Sport	Years Since Retirement	Validated Tool Used	Key Finding	Study Limitations	Study Quality [17]
Gouttebarge, Aoki and Kerkhoffs (2016) [18] Cross-sectional analysis	Belgium, Chile, Finland, France, Japan, Norway, Paraguay, Peru, Spain, Sweden, Switzerland	Male; *n* = 219 Age: 35.0 ± 6.9 Soccer (Football)	4.4 ± 3.6	4 x adverse nutrition statements (validated)	65% had adverse nutrition behaviour. More likely to occur with life events occurring in the last 6 months.	Lack of clear validation process or statistics	66%
Gouttebarge et al. (2017) [19] Cohort study	Netherlands	Male; *n* = 138 Female; *n* = 143 Age: 50.7 ± 15.1 Non-specified Olympic sports	20.2 ± 15.4	4 x adverse nutrition statements (validated)	More likely to have ED symptoms when had athletic injury, career dissatisfaction, inadequate social support, significant life events or surgery	Lack of clear validation process or statistics, insignificant statistical power	66%
Gouttebarge, Frings-Dresen, and Sluiter (2015) [20] Observational study	Australia, Ireland, Netherlands, New Zealand, Scotland, USA	Male; *n* = 104 Age: 36.0 ± 5 Soccer (Football)	5 ± 3	4 x adverse nutrition statements (validated)	42% had adverse nutrition behaviour compared to 26% of current players.	Lack of clear validation process or statistics	33%
Gouttebarge, Kerkhoffs, and Lambert (2016) [21] Cross-sectional analysis	France, Ireland, South Africa	Male; *n* = 295 Age: 38 ± 6 Rugby Union	8 ± 5 Range: 1–25	4 x adverse nutrition statements (validated)	61.9% had adverse nutrition behaviour, made more likely with significant life events and career dissatisfaction	Lack of clear validation process or statistics	66%
Kerr, DeFreese and Marshall (2014) [22] Cross-sectional study	USA	Male; *n* = 376 Female; *n* = 421 Age: 22–51 Various NCAA Division 1 Sports	Unknown	No	5.8% (*n* = 46) had uncontrolled eating, higher than the USA general population	Un-validated tool with no confounding factors	0%
Marquet, et al. (2013) [23] Longitudinal	France	Female; *n* = 20 Male; *n* = 84 Age: unspecified Rowing, Wrestling, Boxing, Judo	Unspecified	Eating Attitudes Test (EAT-26)	Those who engaged in diets in their athletic career were likely to have high ‘dietary restraint’ into retirement	Sampling rationale limited	66%
O’Connor, Lewis, Kirchner and Cook (1996) [24] Cross-sectional	USA	Female; *n* = 22 Age: 36.6 ± 3.8 Gymnastics	~15 years	Eating Disorder Inventory 2 (EDI-2)	Retired athletes were less likely to have body dissatisfaction compared to controls	Sampling rationale limited	66%
Stephan, Torregrosa and Sanchez (2007) [25] Cross-sectional	France	Males; *n* = 46 Females; *n* = 23 Age: 34.88 ± 4.82 Various Sports	5.15 ± 3.28	No	Those who experienced difficulties related to the body changes were negatively related to self-esteem, physical self-worth, physical condition, sports competence and attractiveness.	Lack of validated tool and no mention of sampling rationale	33%

**Table 3 nutrients-11-01395-t003:** Eligible mixed-methods and multi-method studies.

Author, Year, Methodology, Methods	Country	Participant Characteristics	Years Since Retirement	Validated Tool Used	Key Quantitative Finding	Key Themes	Study Quality [17]
Greenleaf (2002) [26] Cross-sectional, thematic analysis	USA	Female; *n* = 6 Age: 23–31 (M = 26) Various Sports	1–5 years	Figure Rating Scale (FRS)	Nil significant findings (The FRS served as an additional source of complementary information)	(1) Factors influencing body image: uniforms, teammates, appearance, fitness and coaches, (2) comparison to previous body, (3) social body ideal vs. athletic ideal	33%
Papathomas, Petrie and Plateau (2018) [27] Cross-sectional, interpretivist analysis, multimethod	USA	Female; *n* = 218 Age: 25.72 ± 1.19 Gymnastics, Swimming	2–6 years	No	Years since retirement was unrelated to weight status, satisfaction and control. 55% dissatisfied with weight, 59.6% trying to lose weight.	(1) Move toward the feminine ideal, (2) feeling fat, flabby and ashamed, (3) a continued commitment to a former self, 4) conflicting ideals: the retired female athlete paradox	66%
Plateau, Petrie, and Papathomas (2017) [28] Cross-sectional, inductive analysis, multimethod	USA	Female; *n* = 218 Age: 25.72 ± 1.19 Gymnastics, Swimming	2–6 years	No	Athletes expressed concern on changing body shape and weight with reduced exercise.	(1) Finding new meanings in exercise (2) Negotiating exercise independence (3) Repositioning exercise in a broader life context	66%
Stephan and Bilard (2003) [29] Longitudinal, thematic analysis, multimethod	France	Female; *n* = 8 Male; *n* = 8 Age: 30.6 ± 3.7 Various Sports	1.5 months and 4.5months	Body Image Questionnaire (BIQ)	Decrease in body satisfaction between 1.5–5 months after retirement.	(1) Weight gain and uncertainty about physical capacities, (2) awareness of physical deterioration, (3) unpleasant somatic symptoms, (4) decrease in social recognition	66%

**Table 4 nutrients-11-01395-t004:** Eligible qualitative studies.

Author, Year, Methodology, Methods	Country	Participant Characteristics, Sport	Years Since Retirement	Objective or Research Question	Key Theme	Study Quality [17]
Cooper and Winter (2017) [30] Interpretative phenomenological analysis	USA	Male; *n* = 2 Female; *n* = 4 Age: 26.4 ± 6.3 Swimming	4.8 ± 6.1	How have DE patterns developed in relation to their competitive performance and how have these patterns changed or persisted with the influence of sport retirement?	(1) Pressures unique to swimming, (2) transition to eating pattern awareness, (3) maintaining ideal eating patterns in retirement	100%
Kerr and Dacyshyn (2000) [31] Inductive analysis	Canada	Female; *n* = 7 Age: 16–22 Gymnastics	0.5–5 years	To enhance understanding of the retirement experiences of elite female gymnasts	(1) Retirement phases; nowhere land, new beginnings, transition process	66%
Plateau, Petrie and Papathomas (2017) [32] Inductive and deductive thematic analysis	USA	Female; *n* = 218 Age: 25.72 ± 1.19 Swimming, Diving, Gymnastics	2–6 years	To explore retired female collegiate athletes’ eating practices in the context of the intuitive eating framework	(1) Permission to eat, (2) recognising internal hunger and satiety cues, (3) eating to meet physical and nutritional needs	17%
Stirling, Cruz, and Kerr (2012) [33] Thematic analysis	USA	Female; *n* = 8 Age: 19.88 ± 2.64 Rhythmic Gymnastics	2.75 ± 1.58	To examine retired rhythmic gymnasts’ perceptions of the influence of retirement on their body satisfaction and weight control behaviours	(1) Increased body dissatisfaction, (2) guilt around weight gain, loss of muscle mass and eating habits	33%

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
