# Peer review of "Retired Athletes and the Intersection of Food and Body: A Systematic Literature Review Exploring Compensatory Behaviours and Body Change"

_nutrients, 2019, doi:10.3390/nu11061395_

Round 1
Reviewer 1 Report
Dear Authors,
Congratulation for this interesting and easy to read review, that we are convinced that will be of interest for sport and sport science specialist.
The few points we would like to let on your consideration for the final version of this revision is that, on our opinion, the beginning of your discussion do not completely reflect your results. You write: “The key drivers of poor relationship with food and body in retired athletes are synthesized throughout this review and include 1) continuing athletic identity; 2) establishing unrealistic expectations related to nutritional and body composition changes, and 3) athletes who have significant energy balance changes or significant body composition shifts away from societal body ideals.” What about career dissatisfactions, sports of high energy consumption, influence of time after retreat that are well identified in your results. Also about this point it is may be of interest to know how you reach those 3 point as key drivers.
In your discussion you refer some specificity for women’s, in your results not, maybe you can highlight some differences between sex?
Author Response
Response to Reviewer 1 Comments
Point 1: In our opinion, the beginning of your discussion does not completely reflect your results. You wrote: “The key drivers of poor relationship with food and body in retired athletes are synthesised throughout this review and include 1) continuing athletic identity; 2) establishing unrealistic expectations related to nutritional and body composition changes, and 3) athletes who have significant energy balance changes or significant body composition shifts away from societal body ideals”. What about career dissatisfaction, sports of high energy consumption, influence of time after retreat that are well identified in your results. Also about this point it is may be of interest to know how you reach those 3 point as key drivers.
Response 1: Thank you for your advice. Career dissatisfaction and time since retirement have been taken on as feedback and added as a 4th key driver. Energy consumption was referred to in the 3rd driver, in terms of ‘significant energy balance changes’. These resulting drivers have further been explained in a sentence as follows: ‘These key drivers have been highlighted as the most common and pertinent factors across all appraised studies.’
Point 2: In your discussion you refer some specificity for women’s, in your results not, maybe you can highlight some differences between sex?
Response 2: Additional information on this has now been included in lines 338-340 explaining further limitations around gender differentiation in the included studies of the review, “Limitations of the previous literature was that gender was not often differentiated between in outcome measures. Future studies need to be able to differentiate between gender as compensatory behaviours have been indicated to vary widely.”
Reviewer 2 Report
The following review from Buckley et. al. synthesized 16 manuscripts using PRISMA to identify themes across disordered eating behaviors affecting elite athletes after retirement. The authors described well defined inclusion criteria: English-language peer-reviewed journals, participants included male or female retired athletes, studies referred to eating behaviors, body image/dissatisfaction, and the methods utilized are clear and concise. The authors identify three overarching themes throughout the existing literature: body dissatisfaction and grief, disordered eating and compensation, and long-term influence of sporting culture. While the Included studies had varying outcomes, they were indications of disordered eating, eating disorders, maladaptive food behaviors, restriction, rigid food control, body image issues, weight control, body control, compensatory exercise, negative self-image, and dysmorphia as related to elite athletes during retirement. The authors conceptualize these three themes as Athletic Body Transition, and identify three compensatory/maladaptive behaviors including compensatory exercise, restrictive eating, and binge eating. Overall, the manuscript is clear, well-written and indicates that more education and support is necessary for retiring athletes as they undergo this critical transition. The manuscript would benefit from the following suggestions.
Major comments
The authors propose Athletic Body Transition, but it is unclear how this really differs from body dysmorphic disorder other than the circumstance where elite athletes are experiencing symptoms as a result of retirement. A few sentences throughout describing how this experience is different from BDD is needed.
A more rigorous discussion on how the psychopathology of Athletic Body Transition differs from a non-elite individual who experiences changes in weight/body composition is needed, or the references to psychological state (i.e., negative affect) should be removed.
The authors methods are thorough and well-described, but inclusion of a critical database PSycINFO via Proquest might provide additional studies to the review that help bolster the mental health aspects surrounding Athletic Body Transition.
Lines 264-270 provide a juxtaposition of Athletic Body Transition being sport-dependent. This type of discussion should be woven throughout the manuscript, including the Introduction. For example, lines 190-194 describing the study by Connor et. al. found that retired college gymnasts reported lower body dissatisfaction compared to age-matched college students. This finding seems to be highly dependent on the sport of gymnastics.
Minor comments
Lines 38-40 require citation.
Some minor spelling/ wording mistakes need to be corrected (e.g., “Three these arose including…” in the abstract)
Author Response
Response to Reviewer 2 Comments
Point 1: The authors propose Athletic Body Transition, but it is unclear how this really differs from body dysmorphic disorder other than the circumstance where elite athletes are experiencing symptoms as a result of retirement. A few sentences throughout describing how this experience is different from BDD is needed.
Response 1: Thank you for your feedback. Additional information to further differentiate between BDD and the subclinical and transient process of ‘Athletic Body Transition’ has now been added in lines: 281-285 and states: “This concept is different to body dysmorphic disorder as it is a unique transition known to athletes in having acute body composition changes and in some instances changing between body ideals. This process is often transient, largely subclinical and is not defined by an individual seeing their body differently, rather to do with the acceptance of body changes.”
Point 2: A more rigorous discussion on how the psychopathology of Athletic Body Transition differs from a non-elite individual who experiences changes in weight/body composition is needed, or the references to psychological state (i.e. negative affect) should be removed.
Response 2: As this review comes from a primarily dietary context rather than a psychopathological context we have included a sentence in the introduction defining the aim with more clarity. This has been included in lines 74-75: “Exploring the psychopathology and clinical diagnostics of retired athletes extends beyond the scope of this paper.” In addition to this, the differentiation is made in the above response in lines 281-285.
Point 3: The authors methods are thorough and well described, but inclusion of a critical database PsychINFO via Proguest might provide additional studies to the review that help bolster the mental health aspects surrounding Athletic Body Transition.
Response 3: Thank you for this feedback which will be taken on for any future reviews. We added an additional sentence (Lines 88-89) in the methods to explain that the primary outcome was diet and secondary outcome was body. Studies were excluded if they only referred to body without food/dietary behaviours. This was not made clear initially and as such can see how there was some confusion regarding the mental health aspect of the review paper. Table 1: PICOS Criteria has been amended to indicate these primary and secondary outcome measures.
Point 4: Lines 264-270 provide juxtaposition of Athletic Body Transition being sport dependent. This type of discussion should be woven throughout the manuscript, including the introduction. For example, lines 190-194 describing the study by Connor et al., found that retired college gymnasts reported lower body dissatisfaction compared to age matched college students. This finding seems to be highly dependent on the sport of gymnastics.
Response 4: Thank you for this feedback, we have hypothesised that it is not necessarily the sport, but the ‘sporting culture’ and/or the ‘athletic body ideal pressure’ i.e. the pressures the type of sport put on the athlete for the ‘idealised’ body composition that influence these measures as seen in the O’Connor et al. study. We argue in the piece that what is seen in the gymnasts is described in female swimmers too as per the ‘athletic body ideal pressure’ being the primary factor influencing body acceptance. Thus, the rationale as to why ‘sport type’ is not necessarily referred to throughout.
Point 5: Lines 38-40 require citation.
Response 5: This sentence has now been cited.
Point 6: Some minor spelling/wording mistakes need to be corrected (e.g., ‘three these arose including…” in the abstract)
Response 6: Spelling mistakes have been amended in the abstract, including “Three themes arose including…” and “The concept of Athletic Body Transition is defined, exploring how a lack of body acceptance may lead to maladaptive behaviours related to food, exercise, and body arising in this transitory period”.
Round 2
Reviewer 2 Report
All concerns have been addressed.